# Vapor sublimation and deposition to build porous particles and composites

Hsing-Ying Tung[1], Zhen-Yu Guan[1], Ting-Yu Liu[2] & Hsien-Yeh Chen [1]

The vapor deposition of polymers on regular stationary substrates is widely known to form uniform thin films. Here we report porous polymer particles with sizes controllable down to the nanometer scale can be produced using a fabrication process based on chemical vapor deposition (CVD) on a dynamic substrate, i.e., sublimating ice particles. The results indicate that the vapor deposition of a polymer is directed by the sublimation process; instead of forming a thin film polymer, the deposited polymers replicated the size and shape of the ice particle. Defined size and porosity of the polymer particles are controllable with respect to varying the processing time. Extendable applications are shown to install multiple functional sites on the particles in one step and to localize metals/oxides forming composite particles. In addition, one fabrication cycle requires approximately 60 min to complete, and potential scaling up the production of the porous particles is manageable.

[1] Department of Chemical Engineering, National Taiwan University, Taipei 10617, Taiwan. [2] Department of Materials Engineering, Ming Chi University of Technology, New Taipei City 24301, Taiwan. Correspondence and requests for materials should be addressed to H.-Y.C. (email: hsychen@ntu.edu.tw)

Porous micro- and nanosized particles have been discovered and demonstrated to be critical tools for applications ranging from conventional particulate technologies[1–3] to modern nanomedicine[4–7] mainly because of their interconnected pore structures, large surface areas, small pore sizes, and low densities. In addition, modifications to introduce functional groups on such particles enable advanced tasks or further conjugation with other functional molecules.[8,9] Porous particles can be synthesized using various porogen materials or analogous approaches during the fabrication process, including precipitation, microfluidic, dispersion, or suspension polymerizations, which are based on the phase separation of the porogen material and the crosslinked particle network.[10] Current challenges for this process include the extensive knowledge required about the material's phase transition, the difficulty of isolating the material from the dispersion medium, the poorly controlled, low efficiency production parameters, the associated high cost for mass production, the particle aggregation that can potentially occur during the modification and installation of functional groups, and, finally, the fact that these processes are limited to solution-based approaches. In the present report, we introduce an approach to produce poly-*para*-xylylene particles with flexible size control in the micrometer (<100 μm) and nanometer (50–900 nm) scales and to introduce pore structures (10–30 nm) in such particles. With the introduced fabrication process, a vapor-phase particle synthesis was performed without requiring a sophisticated supersaturation/condensation process for the vapor mixture,[11] which is difficult to control; instead, a regular vapor deposition process on a sublimating substrate was established to produce micro-/nanosized particles. The fabrication process uses two simultaneously occurring and competing processes, sublimation and deposition,[12] to form the porous particles, and the process differs considerably from the conventional freeze-casting approach through introducing a dynamic foreign body of deposited vapor substance into the system to build the desired materials compared to the static green body[13,14] used in freeze-casting. The fabrication process exploits ice particles as templates,

which are produced by simply spraying water droplets on a superhydrophobic surface (water contact angle of approximately 152°), followed by a freezing step to transform the water droplets into the corresponding ice particle templates. For the demonstration, the construction of the porous particles is thus performed via vapor-phase poly-*para*-xylylene deposition on the ice templates, and the construction occurs at the dynamic vapor−solid interface (Fig. 1a) where the space vacated by the sublimating ice is seamlessly occupied by the vapor-phase deposition of poly-*para*-xylylenes.

The reported sublimation/deposition process results in the construction of polymer (poly-*para*-xylylene) particles with a final architecture that is a replica of the ice particle template, i.e., the same dimensions as the ice particles, and in addition, the particle size is controllable by a stage-wise, timed sublimation/deposition process, and pore structures are formed during the deposition/sublimation competition for volume space. Extensions of exploiting metals/oxides-incorporated ice particles and the multifunctional poly-*para*-xylylenes for the production of particles composites are also demonstrated.

## Results

**Fabrication process using vapor sublimation and deposition.** Water droplets were formed by simply spraying a water mist on a superhydrophobic surface, showing the static, advancing, and receding water contact angles were $152.0° ± 0.9°$ (unless otherwise stated, the values represent the mean ± S.D. of six independent measurements), $154.1° ± 0.6°$, and $150.2° ± 0.8°$, respectively, mimicking the Lotus Effect that occurs after rain falls on a lotus leaf. Theoretically, such a superhydrophobic surface can be created using a variety of different techniques, which have been reviewed elsewhere.[15] As indicated in Fig. 1b, the created droplets were in the range of 50 μm size with a size distribution of ±15 μm. More uniform control of the droplet size can also be produced by approaches reported elsewhere,[16,17] and a demonstration of uniformly dispensed droplets from a

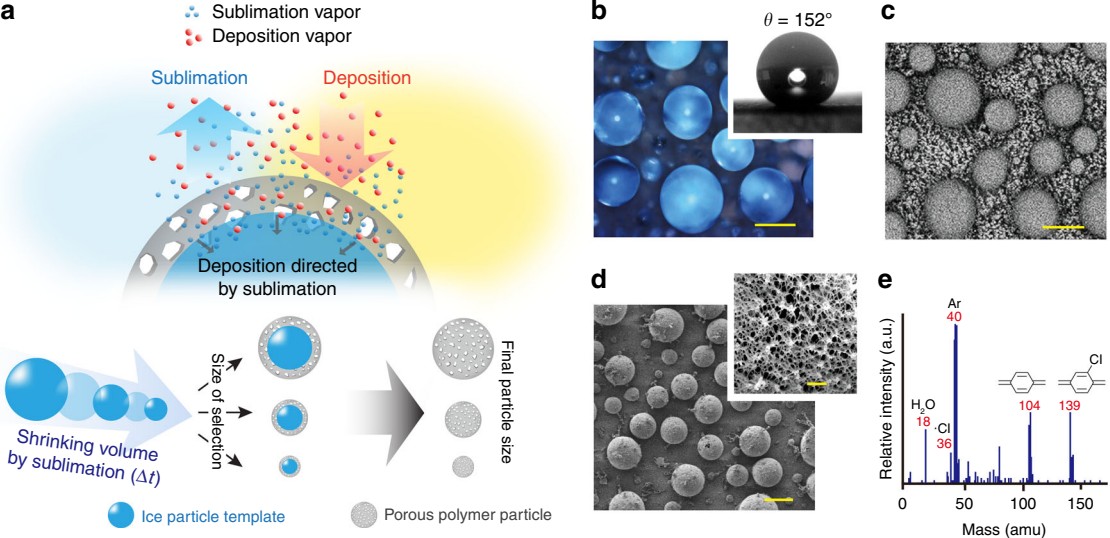

**Fig. 1** Fabrication via sublimation-directed vapor deposition. **a** Schematic illustration of the deposition/sublimation process used to construct micro- and nanosized porous particles. **b** Optical micrograph showing water droplets on a superhydrophobic surface, scale bar: 50 μm; a 152° water contact angle is shown in the top inset. **c** Cryo-SEM micrograph showing the ice particles (templates) created from the water droplets, scale bar: 50 μm. **d** SEM micrograph showing the polychloro-*para*-xylylene particles produced via vapor deposition on the ice particle templates, scale bar: 50 μm. The porous structure with pore size 532.3 ± 89.3 nm formed within the polychloro-*para*-xylylene particle, as indicated in the top inset (scale bar: 5 μm). **e** A mass spectrum recorded during the sublimation/deposition process showing the signals for the sublimated water molecule (18 amu), chloride ion (36 amu), p-quinodimethane (104 amu), and chloro-p-quinodimethane (139 amu)

commercial pipette device was shown in Supplementary Figure 1. The substrates including modified silicon wafer or polytetrafluoroethylene in the study were not reused to avoid potential fouling problem. The resulting droplets-on-a-surface were subsequently frozen in a liquid nitrogen environment to transform the water (liquid) phase into an ice (solid) phase to obtain ice particles (on a surface), as shown in Fig. 1c. Even with the expected volume expansion due to crystallization, the expansion ratio of the ice particles in the micrometer or nanometer ranges was considered to be negligible compared to the original water droplet size. The ice particles served as the deposition template for the vapor deposition and polymerization of poly-*para*-xylylene molecules at 100 mTorr and 4 °C. From a thermodynamic point of view and under such a condition, phase transition of the ice particle occurs naturally from solid phase to the water vapor phase; by contrast, the vapor-phase monomer of the *para*-xylylene undergoes polymerization forming a solid polymer of poly-*para*-xylylene under the same thermodynamic condition. A mass spectrometry-based, real-time gas analyzer was used for characterization during the sublimation/deposition process, and the deposition of a polychloro-*para*-xylylene (chemically inert, highly biocompatible, United States Pharmacopeia Class VI polymer, commercially accessible and named parylene C)[18–21] from the family of poly-*para*-xylylenes, was used first for the demonstration. A sharp peak at 18 amu, which is indicative of water molecules, as well as significant peaks for p-quinodimethane (104 amu) and chloro-p-quinodimethane (139 amu) were detected throughout the entire process, which indicated that sublimation of the water molecules and deposition of the polychloro-*para*-xylylene simultaneously occurred, as shown in Fig. 1e. The deposition mechanism of poly-*para*-xylylenes on a sublimating ice particle is believed to be diffusion-limited, which the ice provides a mobile instead of a stationary substrate and allows limited *para*-xylylene monomers to adsorb onto each surface before it disappears by sublimation. By contrast, an adsorption-limited deposition mechanism was reported for a conventional thin film type poly-*para*-xylylenes deposition, which deposited usually on a stationary substrate.[22,23] The present mechanism has led to the momentary and limited adsorption of the monomers due to the retrograding and sublimating substrate of the ice, and a new direction for the free radical polymerization and the chain propagation is established in association with such a retrograding/sublimating surface (compared to a two-dimensional polymerization and the chain propagation for the thin film poly-*para*-xylylene deposition). The shape of the resultant poly-*para*-xylylene polymer exhibited an approximate replica of the ice particle, as shown in Fig. 1d. The porous structure and the proposed mechanism also show a poignant contrast compared to previous findings examining the formation of encapsulations during vapor deposition of a nonvolatile liquid.[24] The polychloro-*para*-xylylene particles were characterized using Fourier transform infrared spectroscopy (FT-IR), and the recorded spectra (Supplementary Figure 2) showed a characteristic -C-Cl peak at 709−925 cm$^{-1}$ and the -C-H peak in the range 2793−2977 cm$^{-1}$. The positions and intensities of the peaks were consistent with those in the spectra of conventional polychloro-*para*-xylylene thin films. The SEM results indicated that the deposited microsize particles exhibited spherical shapes without any aggregation. The size of the polychloro-*para*-xylylene particles was measured to be 51.1 ± 7.2 μm, showing consistency as compared to the water droplets and the ice particle templates.

## Programmable control of the particle size and voidage.

The rationale for controlling the particle size using the present fabrication technology is further explained here. The fabrication is based on the use of an ice particle template, whose size is limited by only the method used to produce the template, and the size of template determines the replication size of the resulting poly-*para*-xylylene particles. The challenging fabrication of nanosized polychloro-*para*-xylylene particles was chosen to demonstrate the use of the sublimation/deposition fabrication approach. Because depositing polychloro-*para*-xylylene on an ice template using the proposed mechanism results in the formation of an analogous replication architecture based on the ice particle template, using nanosized ice particles or water droplets as the sublimating templates for deposition is a reasonable assumption. However, obtaining nanosized water droplets/ice particles requires high-end equipment with sophisticated controls to manage the presence of nanosized such droplets or particles.[25–27] The equipment or the sophisticated operation procedures have a high capital/operational cost, are not widely available, and do not effectively control the size in the nanometer range. A more versatile and facile approach for obtaining nanosized ice templates and the resulting polychloro-*para*-xylylene nanoparticles was performed herein using a stage-wise, timed sublimation/deposition process. A timed sublimation process was used independently for the ice particle templates, and any possible size could theoretically be used for the fabrication. The sublimating process caused a loss in the mass ($m$) of the ice, and the sublimation kinetic theory was governed by the rate of the mass transfer.

$$h \cdot \Delta T = \frac{\partial}{\partial t} m \cdot \frac{\Delta H_{\text{sub}}}{M_{\text{w}}}, \qquad (1)$$

where $h$ is the heat transfer coefficient of the system and $\Delta T$ is the temperature difference between the operating temperature and the temperature of the sublimating substrate. With the assumption of a constant operation temperature ($\Delta T$ is constant) and properties (heat transfer, $h$, and the latent heat of sublimation, $\Delta H$, are both constant), the equation revealed the change in the ice volume, $\Delta V$, is proportional to the change in the time, $\Delta t$.

During the experiment, the ice particle shrinkage due to sublimation was monitored at 100 mTorr and 4 °C (similar to the deposition conditions in a chemical vapor disposition (CVD) chamber) using a Cryo-SEM. As indicated in Fig. 2a, the ice particles with an initial diameter of 51.2 ± 5.0 μm (7.1 ± 2.0×10$^4$ μm$^3$) shrank to 40.1 ± 4.7 μm (3.5 ± 1.2×10$^{-14}$ μm$^3$) in 60 s, 14.5 ± 2.6 μm (1.7 ± 0.8×10$^3$ μm$^3$) in 120 s, and continued to shrink to 1.1 ± 0.3 μm (8.0 ± 5.9×10$^{-1}$ μm$^3$) after 185 s. The low-quality images from Cyro-SEM were due to the lack of a conducting layer in order for the important observation of sublimation correlation with time variable. The particle volume change closely correlated to Eq. (1), which can be used to predict the particle size from the fabrication. After the timed sublimation process to obtain the desired ice particle template size, the deposition process occurred by introducing the chloro-*para*-xylylene monomer vapor into the system. The vapor polymerized upon deposition to transform the ice template into polychloro-*para*-xylylene particles with the corresponding size (a stage-wise, timed sublimation/deposition process). The predicted size was also further verified for the resulting polychloro-*para*-xylylene nanoparticles, and the programmed sublimation time of 0, 60, 120, and 185 s resulted in the production of polychloro-*para*-xylylene particles with size of 47.3 ± 8.0, 37.8 ± 6.2, 13.7 ± 2.6, and 1.0 ± 0.3 μm, respectively (Fig. 2a, right column); and the size was found comparable to the corresponding initial ice particle template. Demonstration of controllable particle size below submicron range for 418.5 ± 115.3, 141.3 ± 23.4, and 53.7 ± 15.0 nm was also shown in Fig. 2b. In addition, Table 1 has summarized the statistical results and showed the controllability of the ice particles and the resulting polymer particles with the time variable. In separate experiments,

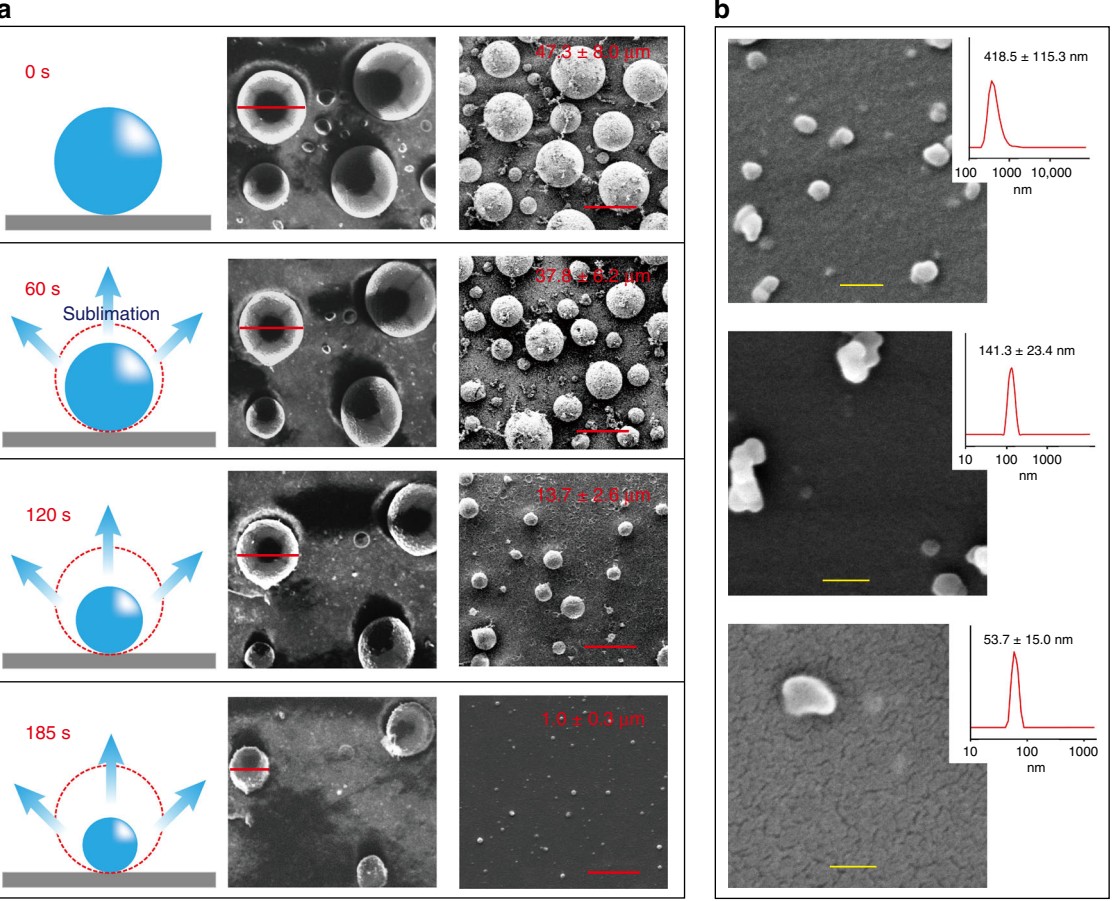

**Fig. 2** Control of particle size. **a** Illustrations of the stage-wise sublimation/deposition process on the left demonstrate the sublimating ice particles and the shrinking volume with respect to the elapsed time. The volume decrease in the ice particles was confirmed by the Cryo-SEM images recorded at the corresponding time shown in the middle column (images were captured at the same location for the same sample, scale bars: 50 μm, 40 μm, 14.5 μm, 1 μm from top down). SEM images of the fabricated polychloro-*para*-xylylene particles with the corresponding size (values represent the mean ± S.D. of six independent measurements) were shown on the right column (samples were prepared separately, scale bars: 50 μm). **b** SEM images and DLS analysis (top insets) of the polychloro-*para*-xylylene particles produced below submicron range for 418.5 ± 115.3 nm (scale bar: 1 μm), 141.3 ± 23.4 nm (scale bar: 150 nm), and 53.7 ± 15.0 nm (scale bar: 50 nm)

---

**Table 1 Statistical results of the diameter ($d_i$, $d_j$) and volume ($V_i$, $V_j$) for ice and polychloro-*para*-xylylene particles obtained with time variable ($t_i$, $t_j$)**

Ice particle

| $i$ | $t_i$ (s) | $d_i$ (μm) | $V_i$ (μm³) | $\Delta V_i$ (μm³)[a] |
|---|---|---|---|---|
| 1 | 0 | 51.2 ± 5.0 | $(7.1 \pm 2.0)\times10^4$ | – |
| 2 | 60 | 40.1 ± 4.7 | $(3.5 \pm 1.2)\times10^4$ | $(3.6 \pm 0.9)\times10^4$ |
| 3 | 120 | 14.5 ± 2.6 | $(1.7 \pm 0.8)\times10^3$ | $(3.3 \pm 1.1)\times10^4$ |
| 4 | 185 | 1.1 ± 0.3 | $(8.0 \pm 5.9)\times10^{-1}$ | $(1.7 \pm 0.9)\times10^3$ |

Polychloro-*para*-xylene particle

| $j$ | $t_j$ (s) | $d_j$ (μm) | $V_j$ (μm³) | $\Delta V_j$ (μm³)[b] |
|---|---|---|---|---|
| 1 | 0 | 47.3 ± 8.0 | $(5.9 \pm 2.9)\times10^4$ | — |
| 2 | 60 | 37.8 ± 6.2 | $(3.0 \pm 1.5)\times10^4$ | $(3.0 \pm 1.4)\times10^4$ |
| 3 | 120 | 13.7 ± 2.6 | $(1.5 \pm 0.7)\times10^3$ | $(2.8 \pm 1.5)\times10^4$ |
| 4 | 185 | 1.0 ± 0.3 | $(6.2 \pm 4.9)\times10^{-1}$ | $(1.5 \pm 0.7)\times10^3$ |
| 5 | 190 | $(418.5 \pm 115.3)\times10^{-3}$ | $(4.4 \pm 3.3)\times10^{-2}$ | $(5.8 \pm 4.1)\times10^{-1}$ |
| 6 | 193 | $(53.7 \pm 15.0)\times10^{-3}$ | $(1.0 \pm 0.9)\times10^{-4}$ | $(4.4 \pm 3.3)\times10^{-2}$ |

[a] $\Delta V_i = V_{i-1} - V_i$
[b] $\Delta V_j = V_{j-1} - V_j$

---

supporting data were shown to flexibly define the particles size from ice particles templates with an initial size of 500 μm, and the data are shown in Supplementary Figure 1. Interestingly, although a spherical shape and contour were expected for the resulting polychloro-*para*-xylylene particles (replicated from the spherical ice particles), an anisotropy in the sphericity was discovered upon the formation of particles with sizes below the submicron level, and this is believed to be due to the monomer deposition and polymerization in random directions within an insufficient deposition time to form homogeneous contours, and/or due to the evolved singularities of ice templates during the sublimation process.[28,29]

In addition, the porous structure of the polychloro-*para*-xylylene particles was further investigated. From a micro- and nanoscopic point of view, the pore structures are formed during the sublimation/deposition process via a combined mechanism of the sublimated gas vapor causing void formation[12] and the polychloro-*para*-xylylene polymer deposition propagating randomly in three dimensions. The same kinetic equation for mass transport in Eq. (1) also describes that the change in the sublimation rate is proportional to the transferred heat gradient, i.e., temperature difference ($\Delta T$). By introducing two new parameters, the operating temperature ($T_o$) and the temperature of the sublimating surface ($T_s$) to the system, the equation with

more explicit temperature parameters can thus be deduced, as shown in Eq. (2).

$$h(T_o - T_s) = \Delta H\left(\frac{dm}{dt}\right) = \Delta H\rho\left(\frac{dV}{dt}\right). \qquad (2)$$

For the described system, the rate of mass loss ($dm/dt$) can also be termed based on the loss of volume ($dV/dt$) with an assumption of constant vapor density ($\rho$); and under the same constant properties and a controlled period of time, the rate of the volume change for the sublimating vapor increased with the increasing operating temperature. During the experiment, the partial pressure of the sublimated water vapor under the varied operating temperatures of 25, 4, and −15 °C was recorded using a real-time gas analyzer, and the results are plotted against the elapsed time in Fig. 3. The data curves demonstrated the predictable pattern from Eq. (2), and the sublimation rate can be calculated. For instance, sublimation for 100 s was 79.1 ± 6.2% faster at 25 °C and 35.3 ± 2.0% faster at 4 °C compared to the sublimation rate at −15 °C. These results again showed a positive correlation with respect to the theoretical prediction from Eq. (2). The sublimation rate prediction was also verified by examining the void space of the resulting polychloro-*para*-xylylene particles. Compared to the theoretical values, the void space increased by 16.7 ± 1.2% (18.8% theoretically) as the operating temperature changed from −15 °C to 4 °C; similarly, a void space increase of 33.5 ± 3.4% was found as the temperature changed from −15 to 25 °C, which was close to the theoretical value of 39.6%. The recorded SEM images of the detailed porous structures with varied sublimation temperatures are included in Supplementary Figure 3. The gradually decreasing water signals also indicated the depletion of the sublimated water vapor, and the curve became a complete depletion line with a slope close to zero, which was also a convenient indication of the sublimation/deposition process completion. For the curve at 25 °C, the complete depletion line occurred at approximately 200 s, and the sublimation time unambiguously supported the size prediction theory in Eq. (1) for 50 μm ice particles.

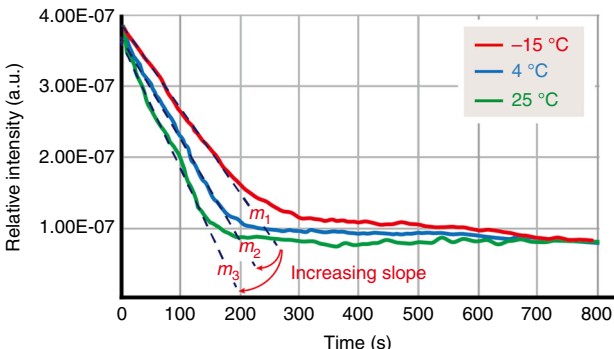

**Fig. 3** Sublimation rate analysis. Real-time analysis of the sublimated water vapor vs. the elapsed time at varied operating temperatures, 25, 4, and −15 °C. The decreasing water vapor intensity correlated to the mass transport rate of the water vapor (sublimation rate) due to the decreasing ice particle volume from sublimation. The sublimation rate was also a function of the operating temperature, and a higher temperature resulted in a higher sublimation rate, which was indicated by a deeper slope in the curve. Based on the slope $m_1$ (−15 °C), $m_2$ (4 °C) increase by 35.3 ± 2.0 %, and $m_3$ (25 °C) increased by 79.1 ± 6.2%. The slopes of the curves gradually fell to a value close to zero, which was indicative of the complete depletion of the water vapor via sublimation. The mean value ± S.D. was calculated from three independent experiments

**Multifunctional particles and composites**. Using the sublimation/deposition process, we demonstrated the production of multifunctional poly-*para*-xylylene particle composites featuring: (i) porous structures with controllable particle sizes and void space, (ii) characteristic metal or oxide properties by compositing the corresponding metal/oxide particles within the structure of the poly-*para*-xylylenes, and (iii) functionalization of the particle composites to provide one or more functional sites for the attachment of a wide range of (bio-)active molecules. The fabrication of these multifunctional particle composites was demonstrated by spraying a water solution containing monodispersed gold nanoparticles (12 nm) on the same superhydrophobic surface to form gold/water droplets, and the transformation of the gold/water droplets to ice particles subsequently occurred using a similar freezing process with liquid nitrogen. The formation of the gold-incorporated polychloro-*para*-xylylene particle composites via the sublimation/deposition process was accomplished using the gold/ice particles from the sublimation of ice as templates and by depositing poly-*para*-xylylene. This left the non-volatile solute, gold, localized in the deposited polymer network and resolved the featured function of (ii). Similar to the sublimation/deposition mechanism described above, the construction of the particle composites also resulted in porous structures within the polymer network, i.e., the feature of (i). Figure 4a shows the TEM characterizations for the resolved gold/ polychloro-*para*-xylylene particles. Separate experiments to incorporate silver or $Fe_3O_4$ nanoparticles in the polychloro-*para*-xylylene particle network were additionally performed to demonstrate the versatility of the fabrication approach (Fig. 4b, c). Noticeably, the concentration of the metal/oxide in one particle composite positively correlated to the concentration of the original metal/oxide solution used when preparing the ice template, and the minimum size of the particle composite was also determined by the concentration. Specifically, the following characteristics were found for the metal/oxide-incorporated polychloro-*para*-xylylene particle composites. First, although the particle size can be predicted using Eq. (1) for the stepwise sublimation/deposition fabrication process, a low critical concentration (LCC) of the metal/oxide solution (for preparing the sublimation template) exists that equals one metal/oxide particle in one produced particle composite, and the minimum size should comply with the size of the metal particle (Fig. 4a). Second, the number of incorporated metal/oxide particles can also be calculated at a concentration higher than the LCC, and the minimum size of the produced particle composite should be close to the collective volume of the confined metal particles (Fig. 4b, c). Third, the shape and contour of the particle composite were found to have a similar spherical anisotropy due to the non-directional polymerization propagation and deposition during the incorporation of the metal particles. Moreover, this fabrication approach overcomes the challenge of potential metal nanocluster formation due to the phase separation that prevents the inclusion of metals within the polymer structure,[30,31] and the approach additionally was able to produce such metal-incorporated poly-*para*-xylylene (polychloro-*para*-xylylene) particle composites in the challenging nanometer range. Finally, in the demonstration, the deposition used a functionalized version of a multicomponent poly-*para*-xylylene that was synthesized using a two-sourced CVD installation, and the deposition occurred by copolymerizing two types of functionalized *para*-xylylene monomers from two independent inlets.[32] The multicomponent *para*-xylylene copolymer containing multiple functional groups was formed in one deposition step. As a proof-of-concept, we choose two functionalized *para*-xylylene monomers, 4-*N*-hydroxysuccinimide-ester-[2,2]para-cyclophane and 4-methyl-propiolate-[2,2]paracyclophane, for the copolymerization deposition process and the demonstration.

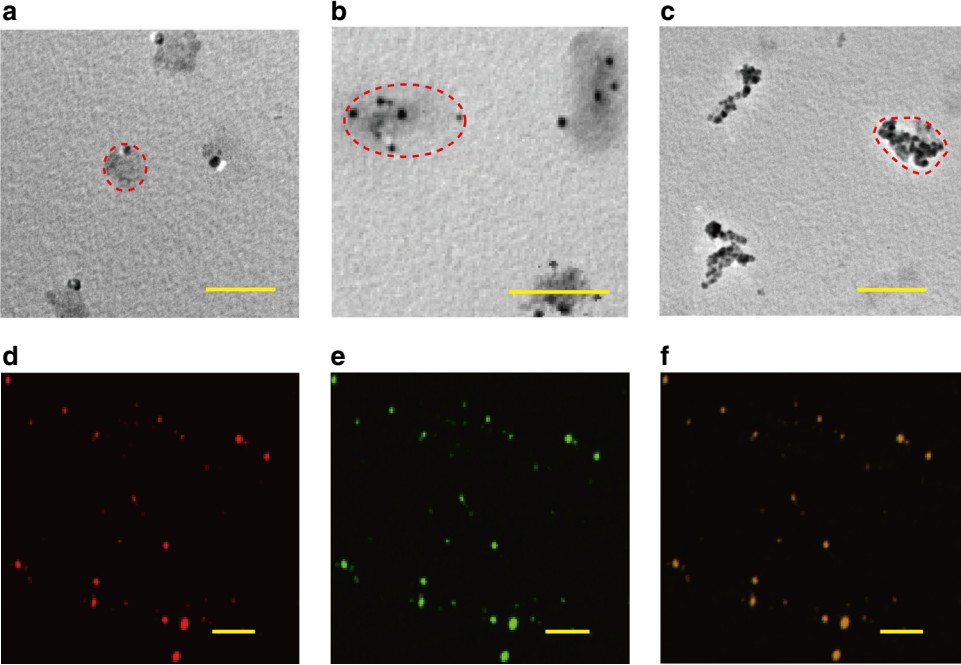

**Fig. 4** Multifunctional particles/composites. TEM images show the production of particle composites by incorporating metals or oxides during the fabrication (sublimation/deposition) process. **a** An average 50.3 ± 4.2 nm particle composite was obtained by incorporating 12 nm gold particles in the composite, scale bars: 100 nm; **b** the incorporation of a number of 6 and 20 nm silver particles resulted in a 189.0 ± 15.3 nm particle composite, scale bars: 200 nm; and **c** a number of approximately 18 and 15 nm $Fe_3O_4$ particles were incorporated and resulted in a 385.7 ± 20.6 nm particle composite, scale bars: 500 nm. Dotted lines were drawn to indicate the contour of the polychloro-*para*-xylylene polymer. The CLSM images show the multifunctional poly-*para*-xylylene copolymer that was used for the incorporation of the $Fe_3O_4$ nanoparticles, and the reactivity of the installed NHS-ester and ester-alkyne groups was confirmed using **d** Alexa Fluor® 555 azide in the red channel (scale bars: 10 μm) and **e** FITC-labeled BSA in the green channel (scale bars: 10 μm). An overlaid image of the red and green channels in **f** indicated the multifunctional presentation and paralleled immobilization of the two fluorescent molecules on the same particle composites (scale bars: 10 μm)

Theoretically, three (or more) sourced depositions and copolymerizations can be realized.[33] The deposition to form a poly[4-N-hydroxysuccinimide-ester-*p*-xylylene-*co*-4-methyl-propiolate-*p*-xylylene-*co*-*p*-xylylene] copolymer on the sublimating $Fe_3O_4$/ice particles can produce the analogous $Fe_3O_4$/poly-*para*-xylylene particle composites and composites with installed functional groups, such as NHS-ester and ester-alkyne groups, which are accessible to the amine/NHS−ester coupling reaction[34] and the azide-alkyne cycloaddition reaction without the requirement of a copper catalyst.[32,35] The use of the multifunctional poly-*para*-xylylene copolymer additionally provided the featured function of (iii) for the proposed, multifunctional poly-*para*-xylylene particle composites. These functionalities were readily available for multitasking, and the production of such multifunctional nanocomposites containing features of (i), (ii), and (iii) can be accomplished in a single step from the same sublimation/deposition process. Similarly, the flexible incorporation of metal (gold and silver) or oxide ($Fe_3O_4$) nanoparticles into the copolymer network was also performed with success, and the TEM and SEM characterizations results are included in Supplementary Figure 4. The availability and orthogonal reactivity of the functional groups of NHS-ester and ester-alkyne groups were examined via conjugation with fluorescein isothiocyanate (FITC)-labeled bovine serum albumin (BSA) and Alexa Fluor® 555 azide using the amine/NHS−ester coupling and copper-free, azide-alkyne cycloaddition reactions, respectively. As shown in Fig. 4d−f, the characterization images from confocal laser scanning microscopy (CLSM) revealed the anticipated fluorescence signals of FITC-labeled BSA in the green channel and Alexa Fluor® 555 azide in the red channel, and the overlaid signals were confined to the same regions where the multifunctional

poly-*para*-xylylene nanoparticles were located. The fabrication approach using the sublimation/deposition process to produce multifunctional poly-*para*-xylylene nanosized particle composites equipped with the multitasking features of (i), (ii), and (iii) required only well-dispersed $Fe_3O_4$ (or other organic or inorganic materials of interest) particles in the solution phase.

## Discussion

The stepwise sublimation/deposition process has demonstrated a facile approach for the fabrication of poly-*para*-xylylene (poly-chloro-*para*-xylylene) particles in the nanometer range with a controllable size. This method alleviates the substantial knowledge required to manage the parameters, including the fluid stability, control of the surfactants, particle size, material chemistry, and post-modification techniques, necessary for conventional fabrication processes. The entire fabrication process requires approximately 60 min to finish a 0.5 g batch (approximately 10 g per day) using a lab scale equipment (40 $cm^3$ chamber), and is compared to a commercial chamber (100 $cm^3$ or larger) for an estimated production capacity of 100 g per day. Considering the challenge of polymer nanoparticle production,[36] scaling up the process for mass production of porous particles is quite manageable.

In conclusion, the proposed fabrication process provides a practical and state-of-the-art synthesis approach for the production of porous nanoparticles and nanocomposites. The results are not limited to poly-*para*-xylylenes and ice (water), as demonstrated in this study. Other vapor-based material deposition and sublimation systems can also be used to construct porous structures, and future research will include discovering the control parameters and the optimal deposition/sublimation properties for

the materials selected. Potential applications for the proposed sublimation and deposition process include use in the fabrication of particles with different combinations of deposition substances onto sublimating substrates under favorable thermodynamic conditions. In addition, the homogeneous incorporation and localization of materials during the proposed fabrication process avoids the concern of phase separation. The reported method enables the production of a versatile particle device with customizable interior properties—for instance, loading of carriers with drugs without relying on hydrophilicity or lipophilicity, loading of sensing particles with tagging materials (e.g., metals, magnetized substances, and isotopes) with varied properties, and achievement of precise concentrations. The proposed mechanism also enables the construction of materials with specialized surface properties, including chemical and/or biochemical functionality, electrostaticity, and other specific affinities of interest.

## Methods

**Ice particle templates and sublimation.** Water droplets (50 μm ± 15 μm in diameter) were formed by spraying deionized water with a mist sprayer (Kingdom, China) on a hydrophobic surface. The surface hydrophobicity was modified on the roughened side of a silicon wafer (Goldeninent Inc., Taiwan) via $C_4F_8$ plasma deposition that was carried out via a radio frequency (13.56 MHz) plasma source (Advanced Energy, USA) with a power of 15 W and a gas flow rate of 50 sccm for 60 s. The hydrophobicity, with an averaged water contact angle of 152°, was confirmed by a contact angle goniometer (First Ten Angstroms, USA). Each measurement was conducted by placing 5 μL of deionized water on the modified surface at three different locations on the same sample surface and was repeated in triplicate for different samples. The water droplets on the surface were frozen in a liquid nitrogen bath to form the solidified ice particle templates. The resulting ice particles were placed in the deposition chamber for the deposition of the poly-para-xylenes under a system pressure of 100 mTorr. During the deposition, varied operation temperatures, including −15, 4, and 25 °C, were used according to the specific experiment for the sublimation of the ice templates. Metal and oxide particles (silver nanoparticle size: 20 nm; gold nanoparticle size: 12 nm; $Fe_3O_4$ nanoparticle size: 10–15 nm) were synthesized following the previous reported procedures.[37–39] The particles were dispersed in deionized water, and the resulting solutions were sprayed on the superhydrophobic surface to form water droplets. This was followed by the same cooling/crystalline process to transform the droplets into the ice particle templates for the CVD deposition.

**Vapor deposition of poly-para-xylenes.** Substituted di-para-xylylene (also known as [2,2]-paracyclophanes) dimers, including dichloro-[2,2]-paracyclophane, 4-N-hydroxysuccinimide ester-[2,2]-paracyclophane, and 4-methyl-propiolate-[2,2] paracyclophane, were used for the CVD. A handmade CVD system capable of performing a single-sourced or dual-sourced synthesis to prepare the poly-para-xylylene monopolymer or copolymer was used in the study. Dichloro-[2,2]-para-cyclophane was purchased commercially (Galxyl C, Galentis, Italy). For the deposition of polychloro-para-xylylene, a temperature of approximately 120 °C was used to vaporize dichloro-[2,2]-paracyclophane, and a carrier gas of 50 sccm of argon was used to transfer the vaporized dichloro-[2,2]-paracyclophane to a pyrolysis area, where the temperature was maintained at 650 °C. The vapor-phase dichloro-[2,2]-paracyclophane was pyrolyzed to form para-xylylene di-radicals (quinodimethanes, monomers) in the pyrolysis zone. Finally, the di-radicals were polymerized on the sublimating ice templates in the deposition chamber at temperatures ranging from −15 °C to 25 °C, resulting in the corresponding polychloro-para-xylylene particles. For the deposition of the multifunctional poly[4-N-hydroxysuccinimide-ester-co-4-methyl-propiolate-p-xylylene-co-p-xylylene] copolymer, the starting materials (dimers), 4-N-hydroxysuccinimide ester-[2,2]-paracyclophane and 4-methyl-propiolate-[2,2]paracyclophane, were used for the CVD copolymerization. The syntheses of the two dimers were conducted following previously reported procedures.[35,40] During the CVD process, a dual-sourced configuration for the CVD system was used for the copolymerization. Specifically, the feeding ratio (molar) of 4-N-hydroxysuccinimide ester-[2,2]-paracyclophane and 4-methyl-propiolate-[2,2]paracyclophane was controlled at 1:1, and pyrolysis temperatures of 700 and 650 °C were used to transform the two dimers into the corresponding quinodimethanes, respectively. The final poly[4-N-hydroxysuccinimide-ester-p-xylylene-co-4-methyl-propiolate-p-xylylene-co-p-xylylene] copolymer was deposited on the sublimating ice templates in the deposition chamber at temperatures ranging from −15 to 25 °C to result in the corresponding multifunctional poly-para-xylylene particles. The deposition rate was regulated at less than 0.5 Å s$^{-1}$ and was monitored using an in situ quartz crystal microbalance analysis (STM-100/MF, Sycon Instruments, USA). The system pressure was controlled at 100 mTorr during the entire CVD (co-)polymerization process.

**Characterizations.** The real-time analysis of the molecular mass was performed using an in situ gas analyzer spectrometer (Hiden Analytical, UK) installed on the homemade CVD system. The mass analysis was performed at $10^{-8}$ Torr with an ionization electron energy of 70 eV and ionization emission current of 20 μA. The detection range for the molecular mass was from 1 amu to 510 amu. FT-IR was performed using an FT-IR 100 spectrometer (PerkinElmer, USA) equipped with a liquid nitrogen-cooled MCT detector. An attenuated total reflectance accessory with a diamond/ZnSe crystal (PIKE Technologies, USA) was used for the characterization of the poly-para-xylylene nanoparticles, and an advanced grazing angle specular reflectance accessory (AGA, PIKE Technologies, USA) was used to characterize the poly-para-xylylene thin films for comparison. The FT-IR spectra were recorded using 128 scans, a 4 cm$^{-1}$ resolution, and a wavenumber from 500 to 4000 cm$^{-1}$. The resulting spectra were corrected for any residual baseline shifts. Images of the ice particle templates were recorded using a Cryo-SEM (Tabletop TM-3000, Hitachi, Japan) with a primary energy of 15 keV and a pressure of 100 mTorr. During the Cryo-SEM analysis, the samples were mounted on the sample holder and cooled by liquid nitrogen. Images of the poly-para-xylylene nanoparticles were obtained using a Nova$^{TM}$ NanoSEM (FEI, USA) with a primary energy of 10 keV and a pressure of $5 \times 10^{-6}$ Torr in the specimen chamber. Samples were prepared by drying the particle suspensions on the silicon wafer and coating them with a conductive layer before the SEM analysis. The TEM analysis was performed using an H-7650 TEM (Hitachi, Japan) at an accelerating voltage of 75 keV, and the samples were prepared by drying the poly-para-xylylene nanoparticle solution on a carbon-deposited copper grid (300 mesh, Electron Microscopy Science, USA). The drying process was conducted using a vacuum oven (DengYng Instruments, Taiwan) at 25 °C. For the particle size measurement, a combination of SEM and DLS was used for the characterizations. In SEM images, the arithmetic mean value of the particle diameter was calculated based on the measurements of lengths along the main axis and perpendicular to the main axis of the particles, and the DLS analysis was performed by using a Zetasizer Nano ZS DLS system (DLS, Malvern, UK) and operated at 25 °C with deionized water as the dispersant. The resulting DLS data were calculated using the non-negative least squares algorithm provided by the built-in Zetasizer software. The fluorescence signals from the immobilized particles were observed using a TCS SP5 CLSM (Leica Microsystems, Germany). An Ar/ArKr laser (wavelength: 488 nm) and an HeNe laser (wavelength: 532 nm) were used to detect the FITC-labeled BSA (detection wavelength: 505–525 nm) and Alexa Fluor® 555 azide (detection wavelength: 580–600 nm).

**Conjugations.** Verification of the reactivity of the two functionalities of the ester-alkyne and N-hydroxysuccinimide esters (NHS-ester) was conducted by reacting fluorescently labeled molecules with the two groups. Specifically, Alexa Fluor® 555 azide (500 μg mL$^{-1}$, Yao-Hong Biotechnology, Taiwan) was used to detect the ester-alkyne group on the multifunctional poly-para-xylylene nanoparticles. A conjugation reaction, an azide-alkyne cycloaddition/click reaction, was expected without a copper catalyst, and the reaction was conducted at 25 °C in an aqueous solution (pH of 7.4) for 2 h. A centrifugation rinsing process (14,000 rpm) using deionized water as a resuspension agent was performed afterwards to remove any unreacted Alexa Fluor® 555 azide. For the NHS-ester group on the multifunctional poly-para-xylylene nanoparticles, the detection was conducted using FITC-labeled BSA (2 mg mL$^{-1}$, Sigma Aldrich, USA), and the conjugation reaction through the amine/NHS−ester coupling reaction was performed at 25 °C in an aqueous solution for 4 h. Finally, the same centrifugation rinsing process was used to remove any unreacted FITC-labeled BSA.

**Data availability.** All the data supporting the findings of this study are available within the article and its supplementary information files and from the corresponding author upon reasonable request.

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

## Acknowledgements

H.-Y.C. gratefully acknowledges support from the Ministry of Science and Technology of Taiwan (MOST 104-2628-E-002-010-MY3) and from the National Taiwan University (104R7745 and NTU-CDP-107L7732). This work is further supported by grants from Advanced Research Center of Green Materials Science and Technology (107L9006 and MOST 107-3017-F-002-001).

## Author contributions

H.-Y.C. conceived the idea of the study. H.-Y.T. and Z.-Y.G. contributed equally to conduct the experiments and the characterizations. H.-Y.T., Z.-Y.G., T.-Y.L., and H.-Y.C. prepared the manuscript and contributed to the discussions of this study.

## Additional information

**Competing interests:** The authors declare no competing interests.

