## [Peer Review File · Nature Communications]

Reviewers' comments:

Reviewer #1 (Remarks to the Author):

This is an interesting paper that demonstrates the use of sublimation and deposition to make polymer nanoparticles. Overall, this paper is interesting but the statistical data is lacking. Although the proof of concept is good, there should be more reproducible data shown as listed below before publication:

- 1) There is a lack of statistics throughout the paper. For example, there are no error bars in the sublimation experiments that show how the ice shrinks with time (lines 160-168). How many times were these experiments done and how reproducible is this data?
- 2) Similarly, is the data in lines 171-175 reproducible? There are no error bars.
- 3) Similarly, are the data in lines 209-220 reproducible?

Reviewer #2 (Remarks to the Author):

After carefully reviewing the manuscript by HY Chen et al, I cannot advise the article for publication in Nature Communications. The article does not have the novelty and extreme importance for the journal. The authors present a synthetic approach of fabrication of porous particles by sublimation of ice particles and polymer deposition. The porous structure formation is suspected to be due to the volume change of ice particles from step-wise sublimation. Although the approach seems to be general and effective, I don't find the mechanism is convincing. No quantitative measurements or characterization are provided to validate their proposed mechanism. Furthermore, freeze-cast method for preparation of porous structures has been around for more than at least one decade. It would be essential that the authors can provide more insights on the physical mechanism in their case related to ice particles. In the current form this article is more for a specialized Journal.

Reviewer #3 (Remarks to the Author):

Tung et al. report the preparation of porous particles via chemical vapor deposition of [2.2]paracyclophanes. So far, most of the work has been focused on continuous films and the

preparation of particle is highly innovative. The manuscript is generally well written and should be published after the following comments are addressed:

- The general quality of the images is low and there are not entirely convincing. The authors may want to consider replacing them with better images.
- The deposition will take a finite time, meanwhile the templating ice particles will be changing. How does this de facto change important design variables such as particle size?
- There should be a statistical analysis of particle sizes as a function of key synthesis parameters.
- There are no direct porosity data. This should be included. Also, a statistical discussion of pore sizes and pore shapes would be critical.
- what is the underlying mechanism for the formation of porous particles? One would expect rather the formation of capsules?
- TEM images should be included
- A brief discussion of utility of these particles would further strengthen the potential impact.

We would like to thank all reviewers of our manuscript for their comments and suggestions for improvement on our manuscript. In the following, we will address all comments and explain our rational and resulting changes to the manuscript in detail. The original statements of the reviewers are shown in plain black, our responses in bold red. For easiness of follow-up, we marked the changes yellow in the main manuscript.

Response to Reviewer 1:

Comments:

This is an interesting paper that demonstrates the use of sublimation and deposition to make polymer nanoparticles. Overall, this paper is interesting but the statistical data is lacking. Although the proof of concept is good, there should be more reproducible data shown as listed below before publication:

We appreciate the comments of the reviewer and are very delighted with the positive feedback.

Q1. There is a lack of statistics throughout the paper. For example, there are no error bars in the sublimation experiments that show how the ice shrinks with time (lines 160-168). How many times were these experiments done and how reproducible is this data?

We appreciate the comments of the reviewer. We have, in the revised manuscript, included additional measurement of the particle size to improve the statistical analysis as suggested by the reviewer; error bars are included for the related analysis throughout the paper, and a summarized **Table 1 is included to demonstrate these results. In addition, details of the statistical analysis are also included in the Materials and Methods in the Supplementary Materials. The additional data are also shown below for the review:**

Table 1. Statistical results of the diameter and volume for ice and polychloro-*para*-xylylene particles obtained with time variable.

time (s)	diameter (μm)	volume (μm^3)	Δ volume (μm^3)
ice particle			
0	51.2 ± 5.0	$(7.1 \pm 2.0) \times 10^4$	60	40.1 ± 4.7	$(3.5 \pm 1.2) \times 10^4$	
120	14.5 ± 2.6	$(1.7 \pm 0.8) \times 10^3$	
185	1.1 ± 0.3	$(8.0 \pm 5.9) \times 10^{-1}$	
polychloro-para-xylylene particle			
185	$(997.0 \pm 139.3) \times 10^{-3}$	$(5.4 \pm 2.2) \times 10^{-1}$	190	$(418.5 \pm 115.3) \times 10^{-3}$	$(4.4 \pm 3.3) \times 10^{-2}$	
192	$(141.3 \pm 23.4) \times 10^{-3}$	$(1.6 \pm 0.7) \times 10^{-3}$	

In Materials and Methods (page 3) in the Supplementary Materials:

.....For the particle size measurement, a combination of SEM and DLS were used for the characterizations. In SEM images, the arithmetic mean value of the particle diameter was calculated based on the measurements of lengths along the main axis and perpendicular to the main axis of the particles, and the DLS analysis was performed by using a Zetasizer Nano ZS DLS system (DLS, Malvern, UK) and operated at 25 °C with deionized water as the dispersant. The resulting DLS data were calculated using the non-negative least squares (NNLS) algorithm provided by the built-in Zetasizer software.

Q2. Similarly, is the data in lines 171-175 reproducible? There are no error bars.

We appreciate the comments of the reviewer. In response to the reviewer's comments, error bars are included for the related analysis throughout the paper (the figures and in the text of the paper), there statistical results are organized and summarized in the additional **Table 1**. These statistical results are showing reproducibility of the fabricated particles, and most importantly, the particle size is predictable and controllable with time variable during the fabrication process.

Q3. Similarly, are the data in lines 209-220 reproducible?

We appreciate the comments of the reviewer. We have, in the revised manuscript, included additional measurement of the pore size, and the error bars are now included for the data in the Supplementary Materials in **Figure S2** and in the text (**page 10**) of the paper.

Response to Reviewer 2:

Comments:

After carefully reviewing the manuscript by HY Chen et al, I cannot advise the article for publication in Nature Communications. The article does not have the novelty and extreme importance for the journal. The authors present a synthetic approach of fabrication of porous particles by sublimation of ice particles and polymer deposition. The porous structure formation is suspected to be due to the volume change of ice particles from step-wise sublimation. Although the approach seems to be general and effective, I don't find the mechanism is convincing. No quantitative measurements or characterization are provided to validate their proposed mechanism. Furthermore, freeze-cast method for preparation of porous structures has been around for more than at least one decade. It would be essential that the authors can provide more insights on the physical mechanism in their case related to ice particles. In the current from this article is more for a specialized Journal.

We appreciate the comments of the reviewer and are delighted with the feedback. The goal of this study was the conception and demonstration of a versatile and generalized particle/composite system based on the use of a unique vapor sublimation and deposition process in which the two competing processes occur at the same time. Evidence was obtained that both the sublimated water vapor and the deposited polymer precursor vapor existed at the same time during the sublimation and deposition process through in situ analysis of the vapor composition during the proposed fabrication process (**Figure 1**). Depletion of the sublimation was also confirmed (**Figure 3**), and the formation of the particle/composite system was observed and characterized (**Figure 1, 2, 4, and Supplementary Materials**). To the best of our knowledge, this unique fabrication process and its mechanism has never been investigated, and the produced particle/composite system is generalized and versatile for use in multiple applications, characteristics which are highly difficult to achieve via conventional methods. In comparison of this newly developed process with the freeze-casting approach, as requested by the reviewer, the proposed process and mechanism differ considerably in several aspects. (i) The freeze-casting technique consists of depositing interventional porogen agents (e.g., solvents) onto existing materials for fabrication, as discussed in the introduction section, and

the process is manipulated by solidification and sublimation of the solvent component. In contrast, the proposed sublimation and deposition process involves the building of materials through a dynamic process using the deposited materials to fill the spaces vacated by the sublimation of ice. (ii) Instead of the stationary “green body” used in the freeze-casting approach, the proposed sublimation and deposition system utilizes a “foreign body” of deposited vapor to build the desired materials. (iii) The mechanism behind freeze-casting is governed by capillary force and osmotic pressure, which separate the solvent phase from the “green body” phase, while these forces do not impact the proposed process. (iv) Finally, the mass transport phenomenon dominates the mechanism for the sublimation and deposition system in that both the sublimated vapor (moving out of the system) and the deposited vapor (moving into the system) exist in a competing fashion, whereas in the freeze-casting system, the only mass transfer is that of solvent escaping from the system. We have also included a discussion comparing the proposed mechanism with conventional approaches and identifying processes in which the new mechanism can be used instead (**page 4**), with relevant citations being included. In addition, we have included a more detailed statistical analysis (also suggested by reviewers) in the revised manuscript and summarized the resulting data in **Table 1** in order to facilitate better reader understanding. We also included an additional discussion providing insights for the use of the particle/composite system and/or the fabrication process in the revision of the manuscript (**page 14**). The additional data are also shown below for the review:

In page 4:

..... The proposed process differs considerably from the conventional freeze-casting approach through introducing a dynamic “foreign body” of deposited vapor substance into the system to build the desired materials compared to the static “green body” used in freeze-casting.

Table 1. Statistical results of the diameter and volume for ice and polychloro-para-xylylene particles obtained with time variable.

time (s)	diameter (μm)	volume (μm^3)	Δ volume (μm^3)
ice particle			
0	51.2 ± 5.0	$(7.1 \pm 2.0) \times 10^4$	 $(3.6 \pm 0.9) \times 10^4$
60	40.1 ± 4.7	$(3.5 \pm 1.2) \times 10^4$	
120	14.5 ± 2.6	$(1.7 \pm 0.8) \times 10^3$	
185	1.1 ± 0.3	$(8.0 \pm 5.9) \times 10^{-1}$	 $(1.7 \pm 0.9) \times 10^3$
polychloro-para-xylylene particle			
185	$(997.0 \pm 139.3) \times 10^{-3}$	$(5.4 \pm 2.2) \times 10^{-1}$	 $(4.9 \pm 1.9) \times 10^{-1}$
190	$(418.5 \pm 115.3) \times 10^{-3}$	$(4.4 \pm 3.3) \times 10^{-2}$	
192	$(141.3 \pm 23.4) \times 10^{-3}$	$(1.6 \pm 0.7) \times 10^{-3}$	

In page 14:

.....Potential applications for the proposed sublimation and deposition process include use in the fabrication of particles with different combinations of deposition substances onto sublimating substrates under favorable thermodynamic conditions. In addition, the homogeneous incorporation and localization of materials during the proposed fabrication process avoids the concern of phase separation. The new method enables the production of a versatile particle device with customizable interior properties—for instance, loading of carriers with drugs without relying on hydrophilicity or lipophilicity, loading of sensing particles with tagging materials (e.g., metals, magnetized substances, and isotopes) with varied properties, and achievement of precise concentrations. The proposed mechanism also enables the construction of materials with specialized surface properties, including chemical and/or biochemical functionality, electrostaticity, and other specific affinities of interest.

Response to Reviewer 3:

Comments:

Tung et al. report the preparation of porous particles via chemical vapor deposition of [2.2]paracyclophanes. So far, most of the work has been focused on continuous films and the preparation of particle is highly innovative. The manuscript is generally well written and should be published after the following comments are addressed:

We appreciate the comments of the reviewer and are very delighted with the positive feedback.

Q1. The general quality of the images is low and there are not entirely convincing. The authors may want to consider replacing them with better images.

We appreciate the comments of the reviewer. We have performed additional experiments and analysis in order to obtain images with better quality, and new images have been replaced for **Figure 2b-d**, **Figure 4a-b**, and **Figure S3d-f** (in Supplementary Materials). On the other hand, the low quality of the 4 images in **Figure 2a** were due to not having the conduction layer (e.g. Au or C layers) when performing SEM analysis, in order for the important observation of sublimation correlation with time variable, and without the interference from the conduction layer. We have also, in the revised manuscript, included an additional discussion to explain the low-quality images. The additional data are also shown below for the review:

Figure 2. Stage-wise sublimation/deposition process. (a) Analysis of the shrinking ice particles via Cryo-SEM. Illustrations on the left demonstrate the sublimating ice particles and the shrinking volume with respect to the elapsed time. The volume decrease in the ice particles was confirmed by the Cryo-SEM images recorded at the corresponding time shown on the right. The images were captured at the same location for the same sample. (b) A customizable particle size was fabricated for the polychloro-*para*-xylylene particles by controlling the sublimation time of the ice particle templates. The SEM images show the selected particle sizes of 997.0 \pm 139.3 nm, 418.5 \pm 115.3 nm, and 141.3 \pm 23.4 nm were fabricated. The DLS analysis is shown in the top insets.

Figure 4. Multifunctional particles/composites. TEM images show the production of particle composites by incorporating metals or oxides during the fabrication (sublimation/deposition) process. (a) An average 50.3 ± 4.2 nm particle composite was obtained by incorporating 12 nm gold particles in the composite; (b) the incorporation of a number of 6 and 20 nm silver particles resulted in a 189.0 ± 15.3 nm particle composite; and (c) a number of approximately 18 and 15 nm Fe_3O_4 particles were incorporated and resulted in a 385.7 ± 20.6 nm particle composite. Dotted lines were drawn to indicate the contour of the polychloro-*para*-xylylene polymer. The CLSM images show the multifunctional poly-*para*-xylylene copolymer that was used for the incorporation of the Fe_3O_4 nanoparticles, and the reactivity of the installed NHS-ester and ester-alkyne groups was confirmed using (d) Alexa Fluor® 555 azide in the red channel and (e) FITC-labeled BSA in the green channel. An overlaid image of the red and green channels in (f) indicated the multifunctional presentation and paralleled immobilization of the two fluorescent molecules on the same particle composites.

Figure S3. TEM and SEM images of particle composites by using a multicomponent poly-*para*-xylylene copolymer to incorporate metals or oxides including (a) gold nanoparticles, (b) Fe₃O₄ nanoparticles, and (c) silver nanoparticles, during the sublimation/deposition fabrication process.

In page 8:

.....The low-quality images from Cyro-SEM were due to the lack of a conducting layer in order for the important observation of sublimation correlation with time variable.

Q2. The deposition will take a finite time, meanwhile the templating ice particles will be changing. How does this de facto change important design variables such as particle size?

In response to the reviewer's comments, the time is indeed an important design variable for the resulting particle size, and the particle volume change (size) closely correlated to deduced equation-(1) in the text in **page 7**, which can be used to predict the particle size from the fabrication process. In addition, such flexible and adjustable time parameter is able to be used for obtaining the desired particle size, and the process is termed a "stage-wise, timed sublimation/deposition process" in the text. We have also, in the revised manuscript, included

additional discussions for the statistical analysis of the particle size by such timed process, as well as to emphasize the correlation of these design variables, in order for the better understanding of readers. The additional discussions are also shown below for the review:

In page 8:

.....As indicated in **Figure 2a**, the ice particles with an initial diameter of $51.2 \pm 5.0 \mu\text{m}$ ($7.1 \pm 2.0 \times 10^4 \mu\text{m}^3$) shrank to $40.1 \pm 4.7 \mu\text{m}$ ($3.5 \pm 1.2 \times 10^{14} \mu\text{m}^3$) in 60 seconds, $14.5 \pm 2.6 \mu\text{m}$ ($1.7 \pm 0.8 \times 10^3 \mu\text{m}^3$) in 120 seconds, and continued to shrink to $1.1 \pm 0.3 \mu\text{m}$ ($8.0 \pm 5.9 \times 10^{-1} \mu\text{m}^3$) after 185 seconds.

.....The particle volume change closely correlated to equation (1), which can be used to predict the particle size from the fabrication.

.....The predicted size was also further verified for the resulting polychloro-*para*-xylylene nanoparticles, and a programmed sublimation time of 185 s resulted in the production of approximately $997.0 \pm 139.3 \text{ nm}$ polychloro-*para*-xylylene particles, 190 s resulted in $418.5 \pm 115.3 \text{ nm}$ particles, and 192 s resulted in $141.3 \pm 23.4 \text{ nm}$ particles, as shown in **Figure 2b-d**. **Table 1** also summarized the statistical results to demonstrate the change of volume (diameter) for the ice particles and the resulting polymer particles with the time variable.

Q3. There should be a statistical analysis of particle sizes as a function of key synthesis parameters.

We appreciate the comments of the reviewer. We have, in the revised manuscript, included a summarized **Table 1** to demonstrate the statistical results of particle size measurement and the relation with time variable, which is the key parameter to predict and control particle size. The additional Table 1 is also shown below for the review:

Table 1. Statistical results of the diameter and volume for ice and polychloro-*para*-xylylene particles obtained with time variable.

time (s)	diameter (μm)	volume (μm^3)	Δ volume (μm^3)
ice particle			
0	51.2 ± 5.0	$(7.1 \pm 2.0) \times 10^4$	60	40.1 ± 4.7	$(3.5 \pm 1.2) \times 10^4$	
120	14.5 ± 2.6	$(1.7 \pm 0.8) \times 10^3$	
185	1.1 ± 0.3	$(8.0 \pm 5.9) \times 10^{-1}$	
polychloro-para-xylylene particle			
185	$(997.0 \pm 139.3) \times 10^{-3}$	$(5.4 \pm 2.2) \times 10^{-1}$	190	$(418.5 \pm 115.3) \times 10^{-3}$	$(4.4 \pm 3.3) \times 10^{-2}$	
192	$(141.3 \pm 23.4) \times 10^{-3}$	$(1.6 \pm 0.7) \times 10^{-3}$	

Q4. There are no direct porosity data. This should be included. Also, a statistical discussion of pore sizes and pore shapes would be critical.

This is an excellent suggestion by the reviewer. In response to the reviewer's comments, the porous images are demonstrated in the top inset of **Figure 1(d)** and in the Supplementary Materials in **Figure S2**. We have also, in the revised manuscript, included more statistical analysis and discussion for these data (in **page 10**), as suggested by the reviewer. These additional data are also shown below for the review:

In page 10:

The sublimation rate prediction was also verified by examining the void space of the resulting polychloro-*para*-xylylene particles. Compared to the theoretical values, the void space increased by 16.7 ± 1.2 % (18.8 % theoretically) as the operating temperature changed from -15 °C to 4 °C; similarly, a void space increase of 33.5 ± 3.4 % was found as the temperature changed from -15 °C to 25 °C, which was close to the theoretical value of 39.6 %. The recorded SEM images of the detailed porous structures with varied sublimation temperatures are included in the Supplementary Materials in **Figure S2**.

Figure S2. Additional SEM images of the pore structures within the polychloro-*para*-xylylene particles. The pore size and void space were positively correlated to the sublimation rate of the ice particle template during the sublimation/deposition fabrication process. (a) Sublimating ice particles at -15 °C resulted in polymer particles with an approximately 50 nm pore size, and (b) a void space of 45.0 ± 3.1 %; 4 °C resulted in 200 nm and 61.7 ± 4.3 %, and (c) 25 °C resulted in 800 nm and 78.5 ± 6.2 %, respectively. The mean value (\pm SD) was calculated from particles in three independent experiments.

Q5. what is the underlying mechanism for the formation of porous particles? One would expect rather the formation of capsules?

The reviewer raises an excellent point. In response to the reviewer's comments, the capsulation would happen when a liquid is non-volatile during that certain thermodynamic condition and during the vapor deposition. The non-volatile liquid provides a stationary solid surface for the vapor deposition to form a thin film layer, and the capsulation of the liquid is the result; such encapsulated liquid was reported from our previous work in Wu et al., *Chemistry of Materials* **2015**, *27*, 7028. While in this work, a sublimating (volatile) ice provides, by contrast, a dynamic surface allowing two competing process of sublimation and deposition occur at each possible location, and the escape of sublimating vapor to result in a porous polymerized network without forming capsulation. We have also, in the revised manuscript in **page 6**, included an additional discussion regarding the possible capsulation issue by comparing to our previous work. The additional discussion is also shown below for the review:

In page 6:

..... The resulting porous structure and the proposed mechanism show a poignant contrast compared to previous findings examining the formation of encapsulations during vapor deposition of a non-volatile liquid.

Q6. TEM images should be included

We appreciate the comments of the reviewer. Additional TEM images as well as better quality images are included/replaced in the revised manuscript in **Figure 4** and in the Supplementary Materials in **Figure S3** as suggested by the reviewer. The additional data are also shown below for the review:

Figure 4. Multifunctional particles/composites. TEM images show the production of particle composites by incorporating metals or oxides during the fabrication (sublimation/deposition) process. (a) An average 50.3 ± 4.2 nm particle composite was obtained by incorporating 12 nm gold particles in the composite; (b) the incorporation of a number of 6 and 20 nm silver particles resulted in a 189.0 ± 15.3 nm particle composite; and (c) a number of approximately 18 and 15 nm Fe_3O_4 particles were incorporated and resulted in a 385.7 ± 20.6 nm particle composite. Dotted lines were drawn to indicate the contour of the polychloro-*para*-xylylene polymer. The CLSM images show the multifunctional poly-*para*-xylylene copolymer that was

used for the incorporation of the Fe₃O₄ nanoparticles, and the reactivity of the installed NHS-ester and ester-alkyne groups was confirmed using (d) Alexa Fluor® 555 azide in the red channel and (e) FITC-labeled BSA in the green channel. An overlaid image of the red and green channels in (f) indicated the multifunctional presentation and paralleled immobilization of the two fluorescent molecules on the same particle composites.

Figure S3. TEM and SEM images of particle composites by using a multicomponent poly-*para*-xylylene copolymer to incorporate metals or oxides including (a) gold nanoparticles, (b) Fe₃O₄ nanoparticles, and (c) silver nanoparticles, during the sublimation/deposition fabrication process.

Q7. A brief discussion of utility of these particles would further strengthen the potential impact.

This is an excellent suggestion by the reviewer. We have, in the revised manuscript in **page 14**, included an additional discussion with respect to the potential impact of using the particles, as suggested by the reviewer. The discussion is also shown below for the review:

In page 14:

....Potential applications for the proposed sublimation and deposition process include use in the fabrication of particles with different combinations of deposition substances onto sublimating substrates under favorable thermodynamic conditions. In addition, the homogeneous incorporation and localization of materials during the proposed fabrication process avoids the concern of phase separation. The new method enables the production of a versatile particle device with customizable interior properties—for instance, loading of carriers with drugs without relying on hydrophilicity or lipophilicity, loading of sensing particles with tagging materials (e.g., metals, magnetized substances, and isotopes) with varied properties, and achievement of precise concentrations. The proposed mechanism also enables the construction of materials with specialized surface properties, including chemical and/or biochemical functionality, electrostaticity, and other specific affinities of interest.

Reviewers' comments:

Reviewer #1 (Remarks to the Author):

In response to the novelty (questions from Reviewer 2), the authors mention that their system has "unique vapor sublimation and deposition process in which the two competing processes occur at the same time." That is not accurate. The Gupta lab at USC has already shown a system for the past few years that has simultaneous sublimation and deposition and the authors need to properly cite this work:

1) "Simultaneous Polymerization and Solid Monomer Deposition for the Fabrication of Polymer Membranes with Dual-Scale Porosity," S. Seidel, P. Kwong, M. Gupta, *Macromolecules*, 2013, 46, 2976.

2) "Solventless Fabrication of Porous-on-Porous Materials," P. Kwong, S. Seidel, M. Gupta, *ACS Applied Materials & Interfaces*, 2013, 5, 9714.

3) "Formation of Porous Polymer Coatings on Complex Substrates Using Vapor Phase Precursors," S. Seidel, G. Dianat, M. Gupta, *Macromolecular Materials and Engineering*, 2016, 4, 371.

Reviewer #2 (Remarks to the Author):

The manuscript states that the size of porous particles is controllable down to the nanometer scale. However, the SEM images show that the water droplets are not uniform in size. How to control the size of water droplets and of ice particle templates on the superhydrophobic substrate? In supporting materials, the authors mentioned that water droplets were around 50 micron. This seems not consistent with the results in the main text.

Related to statistics of the experimental data in Reviewer 1's comments, is the angle of 152° advancing or receding contact angle of water on the surface? What is the hysteresis of the contact angle?

As for the mechanism for the formation of the porous particles, the revised manuscript has not provided convincing argument. In other words, what is the exact reason for the pore formation? How can one know that the shape of ice templates does not evolve during CVD, but remains spherical? The authors mentioned the shape of the particles sometimes is not spherical. The authors have not commented whether freezing singularities in water drop reported in the following papers may be related to their observation: Snoeijer et al Freezing singularities in water drops. *Physics of Fluids* 24, 091102 (2012). Brunet et al. Pointy ice-drops: How water freezes into a singular shape. *Am. J. Phys.* 80, 764 (2012); doi: 10.1119/1.4726201.

The manuscript has not demonstrated how to produce the particles with controlled size, but simply stated that 'scaling up is a simple engineering management matter' or 'is quite manageable'. Overall the ice templates need to form on the surface. It is not simple to me that the ice templates can be easily scaled up, especially with controlled size. Another concern is sustainability of superhydrophobicity of the substrate. Does it remain superhydrophobic after a few rounds of particle production?

On page 6: It is mentioned that the size distribution of the nanoparticles was measured using dynamic light scattering (DLS) and exhibited a polydispersity index of 0.285. But what are the mean sizes of particles from DLS?

On page 10: It is stated that sublimation rate prediction is consistent with their experimental results. However, no direct comparison is given in the manuscript. Is the influence from the film considered for predication of sublimation rate?

A final minor comment: No substantial revision was made in the manuscript. It is a little bit strange that a new author is added at this stage.

Reviewer #3 (Remarks to the Author):

The manuscript has been significantly improved. All of my comments, and as far as I can tell, also the comments of the other reviewers has been addressed. The paper should be published in *Nature Communications*.

We would like to thank all reviewers of our manuscript for their comments and suggestions for improvement on our manuscript. In the following, we will address all comments and explain our rational and resulting changes to the manuscript in detail. The original statements of the reviewers are shown in plain black, our responses in bold red. For easiness of follow-up, we marked the changes yellow in the main manuscript.

Response to Reviewer 1:

Comments:

Q.1. In response to the novelty (questions from Reviewer 2), the authors mention that there system has "unique vapor sublimation and deposition process in which the two competing processes occur at the same time." That is not accurate. The Gupta lab at USC has already shown a system for the past few years that has simultaneous sublimation and deposition and the authors need to properly cite this work:

- 1) "Simultaneous Polymerization and Solid Monomer Deposition for the Fabrication of Polymer Membranes with Dual-Scale Porosity," S. Seidel, P. Kwong, M. Gupta, *Macromolecules*, 2013, 46, 2976.
- 2)"Solventless Fabrication of Porous-on-Porous Materials," P. Kwong, S. Seidel, M. Gupta, *ACS Applied Materials & Interfaces*, 2013, 5, 9714.
- 3) "Formation of Porous Polymer Coatings on Complex Substrates Using Vapor Phase Precursors," S. Seidel, G. Dianat, M. Gupta, *Macromolecular Materials and Engineering*, 2016, 4, 371.

We appreciate the comments of the reviewer and are very delighted with the positive feedback. The pilot study of sublimation and deposition by Gupta's group has been properly cited in the revised manuscript in **page 3**, and in **page 9**.

Response to Reviewer 2:

Comments:

Q.1. The manuscript states that the size of porous particles is controllable down to the nanometer scale. However, the SEM images show that the water droplets are not uniform in size. How to control the size of water droplets and of ice particle templates on the

superhydrophobic substrate? In supporting materials, the authors mentioned that water droplets were around 50 micron. This seems not consistent with the results in the main text.

We appreciate the comments of the reviewer. The water droplets produced by the mist sprayer in the current study are in the size range of 50 μm and with a size distribution of $\pm 15 \mu\text{m}$. We have also discovered an error in the scale bars for **Figure 1 b, c, and d**, the scale bars were meant to describe 50 μm . We have corrected the problems of the error bars, and the results are consistent with the size of ice particles **Figure 2a** and the description for the droplet size in the Supplementary Materials (we thank the reviewer for the correction). For the question about the uniformity of the droplets, there are approaches in the literature can produce more uniform droplets [Dong et al., *ACS Nano* 2013, 7, 10371; Xu et al., *Journal of Micromechanics and Microengineering* 2014, 24, 115011]. The references are properly cited. We have also, in the Supplementary Materials in **Figure S1** (the numbering of figures is subjected to a modification), included additional results of more uniform droplets by using pipette for the demonstration and the comparison; an additional discussion is also included in **page 5** of the revised manuscript. These additional data are also shown below for the review:

Figure S1. Formation of water droplets with uniform size ($508 \pm 6.5 \text{ nm}$) by using a pipette to transfer water liquids on a polytetrafluoroethylene (PTFE) surface.

In page 5:

..... More uniform control of the droplet size can also be produced by approaches reported elsewhere, and a demonstration of uniformly-dispensed droplets from a commercial pipette device was shown in the Supplementary Materials in Figure S1.

Q.2. Related to statistics of the experimental data in Reviewer 1's comments, is the angle of 152 advancing or receding contact angle of water on the surface? What is the hysteresis of the contact angle?

This is an excellent suggestion by the reviewer. We have included additional information (in the revised manuscript in **page 5**) of the water contact angle for the hydrophobic surface. The advancing and receding angles are 154.1 and 150.2°, respectively; and the water contact angle hysteresis is determined to be 3.9°.

Q.3. As for the mechanism for the formation of the porous particles, the revised manuscript has not provided convincing argument. In other words, what is the exact reason for the pore formation? How can one know that the shape of ice templates does not evolve during CVD, but remains spherical? The authors mentioned the shape of the particles sometimes is not spherical. The authors have not commented whether freezing singularities in water drop reported in the following papers may be related to their observation: Snoeijer et al Freezing singularities in water drops. *Physics of Fluids* 24, 091102 (2012). Brunet et al. Pointy ice-drops: How water freezes into a singular shape. *Am. J. Phys.* 80, 764 (2012); doi: 10.1119/1.4726201.

These are excellent suggestions by the reviewer. The pore is formed due to the polymerization at the site of sublimating ice (water vapor), which is similar to the porogen mechanism that has been described [Seidel et al., *Macromolecules* 2013, 46, 2976]; it has been shown to produce porous polymers from partial polymerization and the subsequent sublimation of the solid monomers. The reference is properly cited in the revised manuscript in **page 12**. For the shape of the resulting particles, it is indeed very insightful to include the references suggested by the reviewer. We have also, in the revised manuscript in **page 8-9**, modified the description regarding the explanations of the non-spherical particles formed based on the suggested references. The modifications are also shown below for the review:

..... an anisotropy in the sphericity was discovered upon the formation of particles with sizes below the submicron level, and this is believed to be due to the monomer deposition and polymerization in random directions within an insufficient deposition time to form homogeneous contours, and/or due to the evolved singularities of ice templates during the sublimation process.....

Q.4. The manuscript has not demonstrated how to produce the particles with controlled size, but simply stated that ‘scaling up is a simple engineering management matter’ or ‘is quite manageable’. Overall the ice templates need to form on the surface. It is not simple to me that the ice templates can be easily scaled up, especially with controlled size. Another concern is sustainability of superhydrophobicity of the substrate. Does it remain superhydrophobic after a few rounds of particle production?

In response to the reviewer’s comments, the control of the particle size is achieved by the described “stage-wise, timed sublimation/deposition process” in the study, and the time is an important design variable for obtaining a desirable particle size. With such timed process, any possible size of the water droplets (ice particle templates) could theoretically be used for the fabrication. For instance, popular techniques to produce water droplets include using a pipette (droplet size > 500 μm), automated printing (size of 50 μm – 100 μm), and a mist sprayer (> 50 μm), and the formed ice particle templates can be used for producing the final polymer particles of the same size with only a matter of time. The mass production of the polymer particles from these water droplets/ ice particles (templates) by using the methods above are easy to perform on hydrophobic substrates (e.g. Teflon, or modified surfaces). With respect to the sustainability of the hydrophobicity, we don’t reuse the substrates, considering the substrates and the modification techniques are easily accessible.

Q.5. On page 6: It is mentioned that the size distribution of the nanoparticles was measured using dynamic light scattering (DLS) and exhibited a polydispersity index of 0.285. But what are the mean sizes of particles from DLS?

In response to the reviewer’s comments, the results of polydispersity index from DLS were meant to describe the fabricated polymer particles in **page 8**, however, such information is also found redundant to the newly-included standard deviation information existed for the particle size analysis (measured by DLS, data were included in previous version of the revised manuscript). In order to prevent further confusion, we have removed the polydispersity index data in the new version of the revised manuscript, and we also thank the reviewer for the correction.

Q.6. On page 10: It is stated that sublimation rate prediction is consistent with their experimental results. However, no direct comparison is given in the manuscript. Is the influence from the film considered for predication of sublimation rate?

In response to the reviewer's comments, varied sublimation rate was analyzed and compared in **page 9 – 10**, and also in **Figure 3**. By controlling the environment temperature, the sublimation rate was found increased with increasing temperature, and the amount of sublimated vapor (from the ice particles) was measured in-situ by a real-time gas analyzer.

Q.7. A final minor comment: No substantial revision was made in the manuscript. It is a little bit strange that a new author is added at this stage.

In response to the reviewer's comments, the first author, Hsing-Ying Tung, has graduated from our research group (July 2017, Master degree) and is no longer available to conduct experiments for this study. The new author, Zhen-Yu Guan, is currently a Ph.D. student in our group, who is doing related research as Tung. During the revised manuscript, Guan prepared new particles, performed SEM and TEM characterizations, as well as the statistically analysis of the resulting data. Guan also, during the past, had many other related publications for our research group [Guan et al., *ACS Biomaterials Science & Engineering* 2017, 3, 1815; Guan et al., *Colloids and Surfaces B: Biointerfaces* 2017, 149, 130; Guan et al., *ACS Applied Materials & Interfaces* 2016, 8, 13812; Guan et al., *ACS Applied Materials & Interfaces* 2015, 7, 14431].

Reviewers' comments:

Reviewer #1 (Remarks to the Author):

The authors have answered my questions.

Reviewer #2 (Remarks to the Author):

The response from the authors to some of comments does not get to the point. The comments have not been addressed adequately.

Q1. The manuscript states that the size of porous particles is controllable down to the nanometer scale. The claim of 'nanometer scale' is still not supported.

Q2. Again, the authors even do not give standard derivation from the measurements.

Q4. The authors have not shown how much gram/kilogram of the product, but simply state it is 'easy' to do so.

Reviewer #3 (Remarks to the Author):

Q1: There remains confusion about the particle sizes that can be achieved with this method. To simplify the discussion, I will distinguish between the initial droplet size (i.e., the size at time 0, tens to hundreds of microns) and the apparent droplet size (i.e., the size at the deposition time, down to hundreds of nanometers). Inherent to the approach, the apparent droplet size changes continuously and it is thus somewhat of an overstatement that the final particle size can be controlled, at least this has not yet been demonstrated in the manuscript (see below). In response to the reviewer's comment, there is a new Figure S1 that demonstrates larger initial droplet sizes (500 microns), but it would be helpful to correlate those to various apparent droplet sizes and then show examples of the

resulting polymer particles. There is also an error in this figure, as the (initial) droplet size is reported to be 508 nm.

Q4: Related to above discussion, the authors argue that they can create various initial droplet sizes, and by doing so, they - theoretically - can control the apparent droplet size used to template the polymer particles. While this statement is convincing, it hasn't really been demonstrated in this paper. One convincing way of addressing the reviewer's comment would be to experimentally access different apparent droplet sizes (potentially by using a range of different initial droplet sizes simultaneously) and then demonstrate how this gives access to defined polymer particle sizes that correlate to the corresponding apparent droplet sizes.

The issue of maintaining the superhydrophobic character of the substrate has been sufficiently addressed by the authors. A sentence that the substrates are not reused to avoid fouling should be included in the manuscript.

We would like to thank all reviewers of our manuscript for their comments and suggestions for improvement on our manuscript. In the following, we will address all comments and explain our rational and resulting changes to the manuscript in detail. The original statements of the reviewers are shown in plain black, our responses in bold red. For easiness of follow-up, we marked the changes yellow in the main manuscript.

Response to Reviewer 2:

Comments:

The response from the authors to some of comments does not get to the point. The comments have not been addressed adequately.

Q.1. The manuscript states that the size of porous particles is controllable down to the nanometer scale. The claim of 'nanometer scale' is still not supported.

We appreciate the comments of the reviewer. We have included additional data in the revised manuscript in **Table 1** to better demonstrate the controllability of the resulting particle size. Briefly, an initial droplet size (ice template) of 50 microns was used to produce particles with varied sizes ranging from 50 microns (at time 0) down to submicrons and approximately 140 nanometers (the original version of **Table 1**). Additional data in the revised **Table 1** have provided wider size range of the final polymer particles in microns and tens of nanometer, and their controllable parameter of sublimation/deposition time is also included for the comparison. Images of the final polymer particles with varied sizes are also rearranged in **Figure 2**. In addition, we have taken into the account the suggestions from the reviewer#3 and have performed, in separate experiments, to demonstrate the capability of producing controllable polymer particle from droplets with initial size of approximately 500 μm , and they were able to produce defined polymer particles accordingly, by using the same controllable parameter of sublimation/deposition time. These additional data are included in the supporting information in **Figure S1**. Collectively, the above data are discussed in the revised manuscript in **page 9**.

Q.2. Again, the authors even do not give standard derivation from the measurements.

We appreciate the comments of the reviewer. We have included the information of the standard derivation for these measurements. The advancing and receding angles are $154.1^\circ \pm$

0.6° and 150.2° ± 0.8°, respectively; and the water contact angle hysteresis is determined to be 3.9° ± 0.8°. The information is also included in the revised manuscript in **page 5**.

Q.4. The authors have not shown how much gram/kilogram of the product, but simply state it is 'easy' to do so.

We appreciate the comments of the reviewer. We have included a discussion regarding the production of the polymer particle using the reported technique. Briefly, based on a lab-scaled chamber size of 40 cm³, the batch process can produce 0.5 gram of the particle in 60 mins of operation (10 gram/day), and the process can be scaled up with a commercial chamber (100 cm³ or larger) and with an estimated production capacity of an order more (100 gram/day) than the lab scale. The additional discussion is included in the revised manuscript in **page 4**.

Response to Reviewer 3:

Comments:

Q.1. There remains confusion about the particle sizes that can be achieved with this method. To simplify the discussion, I will distinguish between the initial droplet size (i.e., the size at time 0, tens to hundreds of microns) and the apparent droplet size (i.e., the size at the deposition time, down to hundreds of nanometers). Inherent to the approach, the apparent droplet size changes continuously and it is thus somewhat of an overstatement that the final particle size can be controlled, at least this has not yet been demonstrated in the manuscript (see below). In response to the reviewer's comment, there is a new Figure S1 that demonstrates larger initial droplet sizes (500 microns), but it would be helpful to correlate those to various apparent droplet sizes and then show examples of the resulting polymer particles. There is also an error in this figure, as the (initial) droplet size is reported to be 508 nm.

We appreciate the suggestions of the reviewer to help clarify the confusions. The already included **Table 1** is meant to provide the information about the controllability of the resulting particle size, and we have included more experimental data to improve the quality of the table in order to prevent further confusion. Briefly, an initial droplet size (ice template) of 50 microns was used to produce particles with varied sizes ranging from 50 microns (at time 0) down to submicrons and approximately 140 nanometers (the original version of **Table 1**). Additional data in the revised **Table 1** have provided wider size range of the final polymer particles in microns and tens of nanometer, and their controllable parameter of sublimation/deposition time is also included for the comparison. Images of the final polymer particles with varied sizes are

also rearranged in **Figure 2**. In addition, we have taken into the account the suggestions from the reviewer#3 and have performed, in separate experiments, to demonstrate the capability of producing controllable polymer particle from droplets with initial size of approximately 500 μm , and they were able to produce defined polymer particles accordingly, by using the same controllable parameter of sublimation/deposition time. These additional data are included in the supporting information in **Figure S1**. The error of the droplet size has also been corrected (508 \pm 6.5 μm).

Q.4. Related to above discussion, the authors argue that they can create various initial droplet sizes, and by doing so, they - theoretically - can control the apparent droplet size used to template the polymer particles. While this statement is convincing, it hasn't really been demonstrated in this paper. One convincing way of addressing the reviewer's comment would be to experimentally access different apparent droplet sizes (potentially by using a range of different initial droplet sizes simultaneously) and then demonstrate how this gives access to defined polymer particle sizes that correlate to the corresponding apparent droplet sizes.

Thanks again for the suggestions from the reviewer. We have performed additional experiments to provide more data for demonstration of controllable particle size. The data are included in the revised manuscript in **Figure 2, Table 1**, and the supplementary materials in **Figure S1**.

The issue of maintaining the superhydrophobic character of the substrate has been sufficiently addressed by the authors. A sentence that the substrates are not reused to avoid fouling should be included in the manuscript.

This is an excellent suggestion by the reviewer. We have included the suggested sentence in the revised manuscript in **page 5**.

REVIEWERS' COMMENTS:

Reviewer #3 (Remarks to the Author):

I am satisfied with the revised manuscript. The reviewers have by and large addressed the comments made by the reviewers. The paper should be published in this form.

We would like to thank all reviewers of our manuscript for their comments and suggestions for improvement on our manuscript. In the following, we will address all comments and our responses. The original statements of the reviewers are shown in plain black, our responses in bold red.

Response to Reviewer 3:

Comments:

I am satisfied with the revised manuscript. The reviewers have by and large addressed the comments made by the reviewers. The paper should be published in this form.

We appreciate the comments of the reviewers and are very delighted with the positive feedback.